# Wavelet Feature Maps Compression for Low Bandwidth Convolutional Neural Networks

## Abstract

Quantization is one of the most effective techniques for compressing Convolutional Neural Networks (CNNs), which are known for requiring extensive computational resources. However, aggressive quantization may cause severe degradation in the prediction accuracy of such networks, especially in image-to-image tasks such as semantic segmentation and depth prediction. In this paper, we propose Wavelet Compressed Convolution (WCC)—a novel approach for activation maps compression for $1 \times 1$ convolutions (the workhorse of modern CNNs). WCC achieves compression ratios and computational savings that are equivalent to low bit quantization rates at a relatively minimal loss of accuracy. To this end, we use a hardware-friendly Haar-wavelet transform, known for its effectiveness in image compression, and define the convolution on the compressed activation map. WCC can be utilized with any $1 \times 1$ convolution in an existing network architecture. By combining WCC with light quantization, we show that we achieve compression rates equal to 2-bit and 1-bit with minimal degradation in image-to-image tasks.

## 1 Introduction

Over the past years, Convolutional Neural Networks (CNNs) have brought significant improvement in processing images, video, and audio (LeCun et al., 2015; Krizhevsky et al., 2017). However, CNNs require significant computational and memory costs, which makes the usage of CNNs difficult in applications where computing power is limited, e.g., on edge devices. To address this limitation, several approaches have been proposed to reduce the computational costs of neural networks. Among the popular ones are weight pruning (Han et al., 2015; Guo et al., 2016), architecture search (Howard et al., 2019), and quantization (Li et al., 2017; Banner et al., 2018a). In principle, all these approaches can be applied simultaneously on top of each other to reduce the computational costs of CNNs.

In this work, we focus on quantization. This approach relieves the computational cost of CNNs by quantizing their weights and activation (feature) maps using low numerical precision so that they can be stored and applied as fixed point integers (Hubara et al., 2017; Banner et al., 2018a). In particular, it is common to apply aggressive quantization (less than 4-bit precision) to compress the activation maps (Esser et al., 2020). Yet, it is known that compressing natural images using uniform quantization is sub-optimal. Indeed, applying aggressive quantization in certain CNNs can lead to significant degradation in the accuracy of the network. The impact is especially evident for the image-to-image tasks such as semantic segmentation (Tang et al., 2019) and depth prediction (Lee et al., 2020), where each pixel has to be assigned a value. In a recent work that targets quantized U-Nets (Tang et al., 2019), activations bit rates are kept relatively high (about 8 bits) while the weight bit rates are lower (down to 2 bits). Beyond that, we note that the majority of the quantization works are applied and tested on image classification (Esser et al., 2020; Li et al., 2020, and references therein).

This work aims to improve the compression of the activation maps by introducing Wavelet Compressed Convolution (WCC) layers. These layers utilize the Haar-wavelet transform (Daubechies, 1992) to compress the activation maps before applying convolutions. This wavelet transform and its inverse can be applied efficiently in linear complexity for each channel, using additions and subtractions, thanks to the simplicity of the Haar basis. The main idea of our approach is to keep the same top $k$ entries (in magnitude) of the transformed activations maps with respect to *all channels* (dubbed as joint shrinkage) and perform the convolution in the wavelet domain on the *compressed* signals, saving significant computations. We show that the transform and shrinkage operations

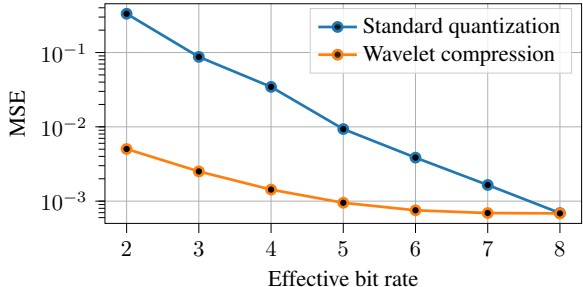

Figure 1: Comparison between standard quantization and our (joint-channel) wavelet compression. The plot shows MSE between the quantized and original activation maps based on a batch of 1000 activation maps from the second hidden layer of MobileNetV3 (small) using the ImageNet data set.

commute with the $1 \times 1$ convolution, the heart of modern CNNs. This procedure is applied along with modest quantization to reduce the computational costs further.

We demonstrate the effectiveness and flexibility of our WCC by applying it in popular network architectures: DeeplabV3plus (Chen et al., 2018) with MobileNetV2 encoder (Sandler et al., 2018) for semantic segmentation, and BTS (Lee et al., 2020) with ResNeXt50 encoder (Xie et al., 2017) for depth prediction. We show that using WCC dramatically improves the results over aggressive quantization for the same compression rates while retaining the baseline network architecture.

## 2 RELATED WORK

**Quantized Neural Networks:** Quantized neural networks have been quite popular recently and are exhibiting impressive progress in the goal for true network compression and efficient CNN deployment. Quantization methods include Zhou et al. (2018); Zhang et al. (2018); Banner et al. (2018b), and in particular Li et al. (2020); Esser et al. (2020); Choi et al. (2019), which show that the clipping parameters—a highly important parameter in quantization schemes—can be learned through optimization. Beyond that, there are more sophisticated methods to improve the mentioned schemes. For example, dynamic quantization schemes utilize different bit allocations at every layer (Dong et al., 2020; Cai & Vasconcelos, 2020; Uhlich et al., 2020). Non-uniform methods can improve the quantization accuracy (Yamamoto, 2021) but require a look-up table, which reduces hardware efficiency. Quantization methods can also be enhanced by combination with pruning (Tung & Mori, 2018) and knowledge distillation for better training (Kim et al., 2019).

The works above focus on image classification. When considering image-to-image tasks like semantic segmentation, networks tend to be more sensitive to quantization of the activations. In the work of AskariHemmat et al. (2019), targeting segmentation of medical images, the lowest bit rate for the activations is 4 bits, and significant degradation in the performance is evident compared to 6 bits. These results are consistent with the work of Tang et al. (2019) that was mentioned earlier, which uses a higher bit rate for the activations than for the weights. The work of Xu et al. (2018) uses weight (only) quantization for medical image segmentation as an attempt to remove noise and not for computational efficiency. The recent work of Liu et al. (2021) shows both a sophisticated post-training quantization scheme and includes fine-tuned semantic segmentation results. Again a significant degradation is observed when going from 6 to 4 bits. One exception is the work of Heinrich et al. (2018) that uses ternary networks (equivalent to 2 bits here) and segments one medical data set relatively well compared to its full precision baseline.

In this work, we focus on the simplest possible quantization scheme: uniform (across all weights and activations), quantization-aware training, with per-layer clipping parameters. That is to ensure hardware compatibility and efficient use of available training data. In principle, one can run our platform regardless of the quantization type and scenario (e.g., non-uniform/dynamic quantization). Also, since we target the compression of the activations, then any method that is focused on the weights (e.g., pruning (Tung & Mori, 2018)) can be combined here as well.

**Wavelet transforms and compression:** Wavelet transforms are widely used in image processing (Porwik & Lisowska, 2004). For example, the JPEG2000 format (Rabbani, 2002) uses the wavelet domain to represent images as highly sparse feature maps. Recently, wavelet transforms have been used to define convolutions and architectures in CNNs for various of imaging tasks. For example, the work of Huang et al. (2017) presents a network architecture for super-resolution, where the wavelet coefficients are predicted and used to reconstruct the high-resolution image. The works of Duan et al. (2017); Williams & Li (2018) use wavelet transforms in place of pooling operators to improve CNNs performance. The former uses dual-tree complex wavelet transform (Kingsbury, 1998), and the latter learns the wavelet basis as part of the network optimization. In these cases, the wavelet transform is not used for compression, but to better preserve information with its low-pass filters.

The work of Wolter et al. (2020) proposes a wavelet-based approach to learn basis functions for the wavelet transform to compresses the weights of linear layers, as opposed to compression of the activation as we apply here. We use the Haar transform (as opposed to a learned one) for its hardware efficiency. Using a different transform (known or learned) is also possible, at the corresponding computational cost. The work of Sun et al. (2021) introduces quantization in the wavelet domain, somewhat similarly to this work. However, the authors suggest to improve the quantization scheme by learning a different clipping parameter per wavelet component, but without the feature shrinkage stage, which is the heart of our approach (we use the same clipping parameter for the whole layer using 8 bits for hardware efficiency). As mentioned before, one can use our WCC together with different types of quantization schemes, in particular including Sun et al. (2021), taking the additional hardware complexity of using different clipping parameter per component into account.

Lastly, Liu et al. (2018) suggest using a modified U-Net architecture for image-to-image translation. There, wavelet transforms are used for down-sampling, and the inverse is used for up-sampling. This work is architecture-specific, and the method can not be easily integrated into other architectures. In contrast, here we propose the WCC layer to easily replace $1 \times 1$ convolutions regardless of the network architecture. Hence our framework is, in principle, also suitable for post-training quantization (Banner et al., 2019), where the data is not available, and the original network is not retrained.

## 3 BACKGROUND

**Quantization-aware training** Quantization schemes can be divided into post- and quantization-aware training schemes. Post training schemes perform model training and model quantization separately, which is most suitable when the training data is not available during the quantization phase (Banner et al., 2019; Nagel et al., 2019; Cai et al., 2020). On the other hand, quantization-aware training schemes are used to adapt the model's weights as an additional training phase. Such schemes do require training data but generally provide better performance. Quantization schemes can also be divided into uniform vs. non-uniform methods, where the latter is more accurate but the former is more hardware friendly (Li et al., 2020; Jung et al., 2019). Lastly, quantization schemes can utilize quantization parameters per channel within each layer or utilize these parameters only per layer (where all the channels share the same parameters). Similar to before, per-channel methods are difficult to exploit in hardware, while per-layer methods are less accurate but are more feasible for deployment on edge devices. This paper focuses on per-layer and uniform quantization-aware training for both weights and activations and aims to improve this with wavelet transforms. Other quantization schemes can be applied within our wavelet compression framework as well.

In quantization-aware training, we set the values of the weights to belong to a small set so that after training, the application of the network can be carried out in integer arithmetic, i.e., activation maps are quantized as well. Even though that we use discontinuous rounding functions throughout the network, quantization-aware training schemes utilize gradient-based methods to optimize the network's weights (Han et al., 2015; Yin et al., 2019). In a nutshell, when training, we iterate on the floating-point values of the weights. During the forward pass, both the weights and activation maps are quantized, while during the backward pass, the Straight Through Estimator (STE) approach is used (Bengio et al., 2013), where we ignore the rounding function, whose exact derivative is zero.

The specific quantization scheme that we use is based on Li et al. (2020). First, the pointwise quantization operator is defined by:

$$q_b(t) = \frac{\text{round}((2^b-1) \cdot t)}{2^b-1}, \tag{1}$$

where $t \in [-1, 1]$ or $t \in [0, 1]$ for signed or unsigned quantization, respectively[1]. The parameter $b$ is the number of bits that are used to represent $t$. During the forward pass, each number undergoes three steps: scale, clip, and round. That is, to get the quantized version of any number, we apply:

$$\text{If signed: } x_b = \alpha q_{b-1}(\text{clip}(\frac{x}{\alpha}, -1, 1)); \quad \text{If unsigned: } x_b = Q_b(x) = \alpha q_b(\text{clip}(\frac{x}{\alpha}, 0, 1)). \quad (2)$$

Here, $x, x_b$ are the real-valued and quantized tensors, and $\alpha$ is the clipping parameter. The parameters $\alpha$ in (2) play an important role in the error generated at each quantization and should be chosen carefully. The works of Li et al. (2020); Esser et al. (2020) introduced an effective gradient-based optimization to find the clipping values $\alpha$ for each layer, again using the STE approximation. This enables the quantized network to be trained in an end-to-end manner with backpropagation. To further improve the optimization, weight normalization is also used before each quantization.

**Haar wavelet transform and compression.** In this section, we describe the Haar-wavelet transform in deep learning language and its usage for compression. See Vyas et al. (2018) for more details on the use of wavelets for image compression. Given an image channel $\mathbf{x}$, the one-level Haar transform can be achieved by a separable 2D stride two convolution with the following kernel:

$$\mathbf{W} = \frac{1}{2} \left[ \begin{bmatrix} 1 & 1 \\ 1 & 1 \end{bmatrix}, \begin{bmatrix} 1 & -1 \\ 1 & -1 \end{bmatrix}, \begin{bmatrix} 1 & 1 \\ -1 & -1 \end{bmatrix}, \begin{bmatrix} 1 & -1 \\ -1 & 1 \end{bmatrix} \right]. \quad (3)$$

These kernels can also be expressed as a composition of separable 1D kernels $[1, 1]/\sqrt{2}$ and $[1, -1]/\sqrt{2}$. The result of the convolution $[\mathbf{y}_1, \mathbf{y}_2, \mathbf{y}_3, \mathbf{y}_4] = \text{Conv}(\mathbf{W}, \mathbf{x})$ has four channels, each of which has half the resolution of $\mathbf{x}$. The leftmost kernel in (3) is an averaging kernel, and the three right kernels are edge-detectors. This, together with the fact that images are piece-wise smooth, leads to relatively sparse images $\mathbf{y}_2, \mathbf{y}_3, \mathbf{y}_4$. Hence, if we retain only the few top-magnitude entries in these vectors, we keep most of the information, as most of the entries we drop are zeros. That is the main idea of wavelet compression. We denote the Haar-wavelet transform by $\mathbf{y} = \text{HWT}(\mathbf{x})$, where $\mathbf{y}$ is defined as the concatenation of the vectors $\mathbf{y}_1, ..., \mathbf{y}_4$ into one. Since the kernels in (3) form an orthonormal basis, applying the inverse transform is obtained by the transposed convolution of (3):

$$\mathbf{x} = \text{iHWT}(\mathbf{y}) = \text{Conv-transposed}(\mathbf{W}, [\mathbf{y}_1, \mathbf{y}_2, \mathbf{y}_3, \mathbf{y}_4]).$$

However, unlike $\mathbf{y}_2, \mathbf{y}_3, \mathbf{y}_4$, the averaged image $\mathbf{y}_1$ is not sparse, and is just the down-sampled original $\mathbf{x}$. Hence, in the multilevel wavelet transform, we apply the kernel in (3) on $\mathbf{y}_1$ to generate further down-sampled sparse channels. For example, a 2-level Haar transform can be summarized as

$$[\mathbf{y}_1^1, \mathbf{y}_2^1, \mathbf{y}_3^1, \mathbf{y}_4^1] = \text{Conv}(\mathbf{W}, \mathbf{x}); \quad [\mathbf{y}_1^2, \mathbf{y}_2^2, \mathbf{y}_3^2, \mathbf{y}_4^2] = \text{Conv}(\mathbf{W}, \mathbf{y}_1^1), \quad (4)$$

and the resulting transformed image with 2-levels can be written as the concatenated vector

$$\mathbf{y} = [\mathbf{y}_1^2, \mathbf{y}_2^2, \mathbf{y}_3^2, \mathbf{y}_4^2, \mathbf{y}_2^1, \mathbf{y}_3^1, \mathbf{y}_4^1] = \text{HWT}(\mathbf{x}). \quad (5)$$

In this work we use 3 levels in all the experiments. To apply compression we define the operator $\mathbf{T}$ that extracts the top $k$ entries in magnitude from the vector $\mathbf{y}$: $\mathbf{y}^{compressed} = \mathbf{T} \cdot \text{HWT}(\mathbf{x})$. To de-compress this vector we first zero-fill $\mathbf{y}^{compressed}$ (multiply with $\mathbf{T}^\top$) and apply the inverse Haar transform, which is the transposed convolutions in the opposite order.

## 4 WAVELET COMPRESSED CONVOLUTION

In this work, we aim to reduce the memory bandwidth and computational cost associated with convolutions performed on intermediate activation maps. To this end, we apply the Haar-wavelet transform to compress the activation maps, in addition to light quantization of 8 bits. Our method is most efficient for scenarios with high-resolution feature maps (i.e., large images, whether in 2D or 3D), where the wavelet compression is most effective. Such cases are mostly encountered in image-to-image tasks like semantic segmentation, depth prediction, image denoising, in-painting, super-resolution, and more. Typically, in such scenarios, the size and memory bandwidth used for the weights are relatively small compared to those used for the features (e.g., point cloud segmentation).

---

[1] In most standard CNNs, the ReLU activation is used; hence the activation feature maps are non-negative and can be quantized using an unsigned scheme. If a different activation function that is not non-negative is used, or, as in our case, the wavelet coefficients are quantized, signed quantization should be used instead.

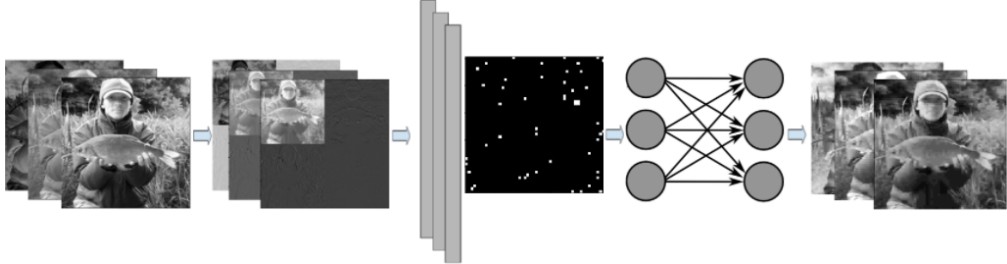

Figure 2: The workflow of WCC. From Left to right: the input channels, their Haar transform, the joint shrinkage of the 2D Haar representation into equal sized 1D vectors and a single bit-map (or a list of indices), the application of the $1 \times 1$ convolution on the 1D vectors, and lastly on the right: the inverse transform back to the spatial domain. Here, the shrinkage ratio is set to $0.1$.

In addition, since we have to predict each pixel, typical architectures do not aggressively down-sample the images, as opposed to multi-class image classification, where activation maps become small, and the number of channels grows towards the end of the network.

**Convolution in the wavelet domain:** Since we focus on computational efficiency, we use the Haar transform, as it is the simplest and most computationally efficient wavelet variant. Indeed, the Haar transform involves binary pooling-like operations, which include only additions, subtractions, and a bit-wise shift. We focus on the compression of fully-coupled $1 \times 1$ convolutions, as these are the workhorse of lightweight and efficient architectures like MobileNet (Sandler et al., 2018), ShuffleNet (Ma et al., 2018), EfficientNet (Tan & Le, 2019), and ResNeXt (Xie et al., 2017). All these modern architectures rely on $1 \times 1$ convolutions in addition to grouped or depthwise spatial convolutions (i.e., with $3 \times 3$ or larger kernels), which comprise a small part of the computational effort in the network—the $1 \times 1$ operations dominate the inference cost (see Appendix B). The main idea of our method is to transform and compress the input using the wavelet transform prior to the $1 \times 1$ convolution, then apply it in the wavelet domain on a fraction of the input size. Since the wavelet compression is applied separately on each channel, it commutes with the $1 \times 1$ convolution. Hence, applying the convolution in the wavelet domain is equivalent to applying it in the spatial domain.

**Joint hard shrinkage:** Prior to the $1 \times 1$ convolution, the spatial wavelet transform is applied, and we get sparse feature maps. Since the different channels result from the same input image propagated through the network, they typically include patterns at similar locations, and hence the sparsity pattern of their wavelet-domain representation is relatively similar. This idea of redundancy in the channel space is exploited in the works of Han et al. (2020); Eliasof et al. (2020); Bae et al. (2021), where part of the channels are used to represent the others. Therefore, we perform a joint shrinkage operation between all the channels, in which we zero and remove the entries with the smallest feature norms across channels, resulting in a compressed representation of the activation maps[2]. The locations of the non-zeros in the original image are kept in a *single* index list or a bit-map for all the channels in the layer, as they are needed for the inverse transform back to the spatial domain. Lastly, we also apply light 8-bit quantization to the transformed images to further improve the compression rate. The weights and wavelet-domain activations are quantized using the symmetric scheme as described in section 3.

More precisely, the advantage of the wavelet transforms is their ability to compress images. Denote the Haar transform matrix as $\mathbf{H}$, i.e., $\mathbf{Hx} = \text{HWT}(\mathbf{x})$. Then, for most natural images we have that

$$\mathbf{x} \approx \mathbf{H}^\top \mathbf{T}^\top \mathbf{THx} \tag{6}$$

where $\mathbf{T}$ is the shrinkage operator described above, and because of its orthogonality, $\mathbf{H}^\top$ is the inverse transform IHWT. Our WCC layer is defined by:

$$\text{WCC}(\mathbf{K}_{1 \times 1}, \mathbf{x}) = \mathbf{H}^\top \mathbf{T}^\top \mathbf{K}_{1 \times 1} \mathbf{THx}, \tag{7}$$

where $\mathbf{K}_{1 \times 1}$ is the learned convolution matrix. The workflow is illustrated in Figure 2. Note that the convolution operates on the compressed domain, hence, if $\mathbf{T}$ can significantly reduce the dimensions of the channels, this leads to major savings in computations.

---

[2]Regardless of the joint sparsity of the channels, some approaches suggest taking the left-upmost part of the wavelet transform for any image, so in principle, the joint sparsity may suffice for a general set of images as well.

Our method aims at compression only. Hence, we show that a $1 \times 1$ convolution kernel can be applied both in the spatial and in the compressed wavelet domain. By its definition, we can write a $1 \times 1$ convolution as a summation over channels. That is:

$$\mathbf{y} = \mathbf{K}_{1 \times 1} \mathbf{x} \Rightarrow \mathbf{y}_i = \sum_j k_{ij} \mathbf{x}_j, \tag{8}$$

where $k_{ij} \in \mathbb{R}$ are the weights of the convolution tensor. Now, suppose we wish to compress the result $\mathbf{y}$, through (6). Because $\mathbf{T}$ and $\mathbf{H}$ are separable and spatial, we get by simple linear arithmetic:

$$\mathbf{y}_i \approx \mathbf{H}^\top \mathbf{T}^\top \mathbf{T} \mathbf{H} (\mathbf{K}_{1 \times 1} \mathbf{x})_i = \mathbf{H}^\top \mathbf{T}^\top \sum_j k_{ij} \mathbf{T} \mathbf{H} \mathbf{x}_j \Rightarrow \mathbf{y} \approx \mathbf{H}^\top \mathbf{T}^\top \mathbf{K}_{1 \times 1} \mathbf{T} \mathbf{H} \mathbf{x}. \tag{9}$$

Hence, the wavelet compression operator, which is known to be highly efficient for natural images, commutes with the $1 \times 1$ convolution, so the latter can be applied on the compressed signal and get the same result as compressing the result directly. This introduces an opportunity to save computations on the one hand and use more accurate compression on the other.

The description above is suited for $1 \times 1$ convolutions only, while many CNN architectures involve spatial convolutions with larger kernels, strides, etc. The main concept is that fully coupled convolutions that mix all channels are expensive, inefficient, and redundant when used with large spatial kernels (Ephrath et al., 2020). The spatial mixing can be obtained using separable convolutions at less cost without losing efficiency (Chen et al., 2018). Since we aim at saving computations, separating the kernels in the architecture is recommended even before discussing any type of lossy compression. Furthermore, separable convolutions (and the Haar transform) can be applied separately in chunks of channels or together as part of the $1 \times 1$ convolution using specialized implementation. Hence the memory bandwidth can be reduced. The $1 \times 1$ convolution, on the other hand, involves all channels at once; hence it is harder to reduce the bandwidth in this case.

### 4.1 COMPUTATIONAL COSTS IN BIT OPERATIONS (BOPS)

To evaluate the computational cost involved in WCC we use the measure of Bit-Operations (BOPs) (Wang et al., 2020; Louizos et al., 2018). First, the number of Multiply-And-Accumulate (MAC) operations in a convolutional layer is given by

$$\text{MAC(conv)} = C_{\text{in}} \cdot C_{\text{out}} \cdot N_W \cdot N_H \cdot K_W \cdot K_H \cdot \frac{1}{S_W \cdot S_H}, \tag{10}$$

where $C_{\text{in}}$ and $C_{\text{out}}$ are the number of input and output channels, $(N_W, N_H)$ is the size of the input, $(K_W, K_H)$ is the size of the kernel, and $(S_W, S_H)$ is the stride value. The BOPs count is then

$$\text{BOPs(conv)} = \text{MAC(conv)} \cdot b_w \cdot b_a, \tag{11}$$

where $b_w$ and $b_a$ denote the number of bits used for weight and activations.

As described in section 3, the Haar transform is separable between the input channels, and can be viewed as four $2 \times 2$ convolutions with stride $(2, 2)$ and binary weights. Hence, the one-level transform requires $4 \cdot C_{\text{in}} \cdot W \cdot H \cdot b_a$ BOPs. The transform can be used with more levels of compression explained in section 3, on down-scaled inputs, resulting in a total of

$$\sum_{l=1}^{L} 4 \cdot C_{\text{in}} \cdot N_W \cdot N_H \cdot \frac{1}{4^{l-1}} \cdot b_a \tag{12}$$

BOPs, where $L$ is the level of compression. Similarly, the inverse-transform result in the same calculation, only with $C_{\text{out}}$ in place of $C_{\text{in}}$. To demonstrate the relatively small cost of the compression, consider a $1 \times 1$ convolution with $C_{\text{in}} = 160$, $C_{\text{out}} = 960$, input size of $(34, 34)$, and quantization $b_w = b_a = 8$ (which is part of a network used later in section 5). This layer costs $11,364M$ BOPs. Using a 3 levels wavelet transform and its inverse for this layer results in $54M$ BOPs, a negligible cost which allows for better compression, as we demonstrate next.

## 5 RESULTS

In this section, we evaluate our WCC layer's performance in image-to-image tasks—semantic segmentation and depth estimation—where the feature maps are relatively large and suitable for wavelet compression. We compare our method to quantization-aware training based on the work of Li et al. (2020), which we also use in our method for 8-bit quantization of the shrunk signals. Our code is implemented in PyTorch (Paszke et al., 2017) based on Torchvision implementations, including the same data augmentation. We ran our experiments on NVIDIA 24GB RTX 3090 GPU. We detail our experimental setup for each section separately.

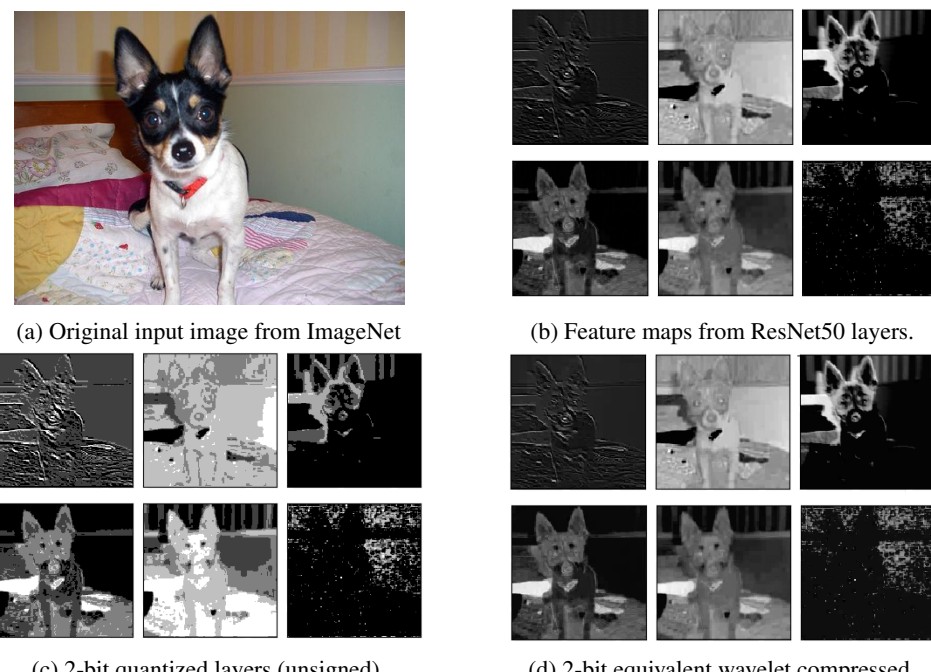

(a) Original input image from ImageNet      (b) Feature maps from ResNet50 layers.

(c) 2-bit quantized layers (unsigned).      (d) 2-bit equivalent wavelet compressed.

Figure 3: Feature maps from layers 2 and 3 (top and bottom triplets, respectively) of a pretrained ResNet50 on ImageNet. The maps are compressed with uniform quantization (2-bit) and wavelet compression (25% shrinkage + 8-bit quantization, equivalent to 2-bit quantization in terms of size). It is clear that wavelet compression loses much less information than aggressive quantization.

## 5.1 QUALITATIVE COMPRESSION ASSESSMENT

Before describing experiments with common datasets and tasks, we first qualitatively demonstrate the advantage of our approach. First, we compare the MSE of standard quantization vs our approach on a feature map from a pretrained MobileNetV3 applied on a batch of 1000 images from the ImageNet dataset (Krizhevsky et al., 2017). We consider an effective bit rate range of $[2, 8]$. The quantization of the shrunk wavelet coefficients is 8 bit, and multiplying it with the compression ratio yields the effective bit rate. E.g., 8 bit and 25% shrinkage is effectively 2 bits. This also agrees with the BOPs measure in Eq. (11). Figure 1 shows the MSE comparison per effective bit rate. One can see that the information loss based on the MSE is more significant in standard quantization. We note that the quantization parameters in this experiment were chosen by an exhaustive search for minimizing the MSE for every dot in the plot. Furthermore, in Figure 3 we can see a visualization comparison of a typical activation map compressed by a standard quantization and our method. It is obvious that the perceptive degradation in the image approximation is more significant in the standard quantization. This is not surprising since wavelet image compression is more advanced than uniform quantization.

## 5.2 SEMANTIC SEGMENTATION

Semantic segmentation is the task of assigning a label to every pixel in the input image. CNNs have shown significant improvements over traditional methods but at a high computational cost. Our model of choice here is the popular DeeplabV3plus (Chen et al., 2018) with MobileNetV2 backbone (Sandler et al., 2018). We evaluated the proposed method on the Cityscapes and Pascal VOC. The Cityscapes dataset (Cordts et al., 2016) contains images of urban street scenes. The images are of size 1024x2048 with pixel-level annotation of 19 classes. During training, we used a random crop of size 768x768 and no crop for the validation set. The Pascal VOC (Everingham et al., 2015) dataset contains images of size 513x513 with pixel-level annotation of 20 foreground object classes and a background class. We augmented the dataset similarly to Chen et al. (2018).

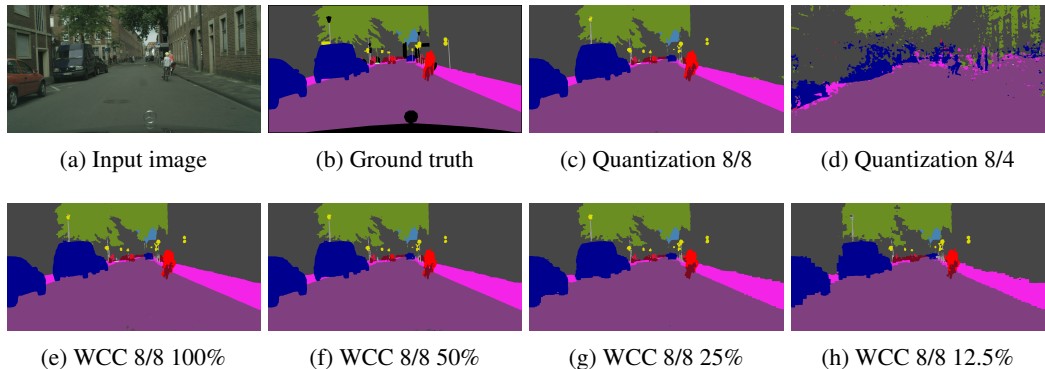

(a) Input image     (b) Ground truth     (c) Quantization 8/8     (d) Quantization 8/4

(e) WCC 8/8 100%     (f) WCC 8/8 50%     (g) WCC 8/8 25%     (h) WCC 8/8 12.5%

Figure 4: Cityscapes segmentation results. All the networks use weight quantization of 8-bits. (a) input image. (b) ground truth. (c), (d) normal quantization with 8- and 4-bits activations respectively. (e)-(h) WCC with 8-bits activations and shrinkage rate of 100%, 50%, 25% and 12.5% respectively.

| Precision (Weight / Activation) | Wavelet shrinkage | Cityscapes BOPs (B) | mIoU | Pascal VOC BOPs (B) | mIoU |
|---|---|---|---|---|---|
| Full precision | None | 36,377 | 0.717 | 4,534 | 0.715 |
| 8bit / 8bit | None | 2,273 | 0.701 | 283 | 0.712 |
| 8bit / 6bit | None | 1,705 | 0.683 | 212 | 0.678 |
| 8bit / 4bit | None | 1,136 | 0.173 | 141 | 0.095 |
| 8bit / 8bit | 100% | 2,292 | 0.697 | 285 | 0.695 |
| 8bit / 8bit | 50% | 1,213 | 0.681 | 150 | 0.675 |
| 8bit / 8bit | 25% | 673 | 0.620 | 82 | 0.611 |
| 8bit / 8bit | 12.5% | 403 | 0.552 | 48 | 0.519 |
| 4bit / 8bit | None | 1,136 | 0.682 | 141 | 0.675 |
| 4bit / 6bit | None | 852 | 0.669 | 106 | 0.657 |
| 4bit / 4bit | None | 568 | 0.190 | 70 | 0.099 |
| 4bit / 8bit | 100% | 1,156 | 0.672 | 144 | 0.678 |
| 4bit / 8bit | 50% | 616 | 0.667 | 76 | 0.661 |
| 4bit / 8bit | 25% | 346 | 0.621 | 42 | 0.583 |
| 4bit / 8bit | 12.5% | 211 | 0.549 | 24 | 0.515 |

Table 1: Validation results for semantic segmentation task using DeepLabV3plus with MobileNetV2 as the backbone. Segmentation performance is measured by mean intersection over union (mIoU).

We used two configurations for the weights—4 and 8 bits—and for each of them we used different compression rates for the activations, both for the standard quantization and our WCC layer. Like before, our WCC layer with a shrinkage rate of 50% has nearly equal BOPs to reducing the activation bits from 8 to 4. Likewise, a shrinkage rate of 25% nearly equals to 2 bits.

The training scheme used is similar to Li et al. (2020). We first trained a network in full precision and then gradually reduced the bit rates (this was used for the WCC as well). For all datasets and model optimization, we used SGD with momentum 0.9, weight decay $10^{-4}$, learning rate decay 0.9, and set the batch size to 16. For Cityscapes, we trained the full precision model for 160 epochs with a base learning rate of 0.1, and each of the retrains was trained for 50 epochs with a base learning rate of 0.01. For Pascal VOC, we trained the full precision model for 50 epochs with a base learning rate of 0.01, and each of the retrains was trained for 25 epochs with a base learning rate of 0.002.

Table 1 shows the performance and BOPs of each model, from which we can see that our model achieves superior performances at high compression configurations, outperforming the 4-bit activation even with compression rates equivalent to 2- and 1-bits. Figure 4 shows a visual example.

| Precision (Weight / Activation) | Wavelet shrinkage | BOPs (B) | AbsRel | RMSE |
|---|---|---|---|---|
| Full precision | None | 156,882 | 0.061 | 2.774 |
| 8bit / 8bit | None | 9,805 | 0.068 | 2.889 |
| 8bit / 8bit | 50% | 4,910 | 0.071 | 3.063 |
| 8bit / 4bit | None | 4,902 | 0.070 | 3.065 |
| 8bit / 8bit | 25%-50% | 4,549 | 0.075 | 3.228 |
| 8bit / 2bit-4bit | None | 4,541 | 0.106 | 4.082 |
| 8bit / 8 bit | 25% | 4,297 | 0.08 | 3.352 |

Table 2: Validation results on BTS, using ResNeXt50 as backbone encoder, measured by absolute relative difference (AbsRel) and root mean squared error (RMSE), in both lower is better.

Comparison to Liu et al. (2021): This work deals with zero-shot quantization and shows results on quantization-aware training (fine-tuned) for Cityscapes, albeit using $256 \times 256$ input images. In their paper, the 8bit/8bit fine-tuned result has a mIoU of 0.613 compared to about 0.7 here, in both the standard and wavelet cases. Our standard 4bit/8bit setting achieves 0.67 vs. 0.6 for 6bit/6bit by Liu et al. (2021). Furthermore, when considering 4bit weights, all our wavelet compression results (up to 12.5%) outperform or are on par with the 4bit/4bit result of 0.56 by Liu et al. (2021).

## 5.3 Monocular Depth Estimation

In this section, we apply our compression technique on Big-to-Small (BTS, Lee et al., 2020), a network architecture for monocular depth estimation—a task of estimating scene depth using a single image. We evaluated the results on the KITTI dataset (Geiger et al., 2013), containing images of autonomous driving scenarios, each of size ~1241x376 pixels. The dataset splits are as in BTS, using the strategy of Eigen et al. (2014). As in BTS, we use a maximum value of 80 meters for prediction when testing, and the performance evaluation is based on the cropping scheme of Garg et al. (2016). We did not modify the initial convolution, the final convolutions, and the depth prediction layer when applying the compression. Using any compression technique on these layers resulted in a significant degradation in the results.

In our experiments, we focus on a ResNeXt50 backbone. The compressed models are fine-tuned from a 8bit/8bit quantized network, trained from scratch. In local planar guidance layers, we use 8bit/4bit quantization instead of our wavelet scheme. When training, we use the same configuration presented in the BTS repository. We use the AdamW optimizer (Loshchilov & Hutter, 2019), a learning rate of 0.00014, a weight decay of 0.001, a batch size of 4 images, and train for 50 epochs.

Table 2 shows a comparison between the standard quantization and WCC for BTS. Using our WCC layer with 50% compression resulted in comparable scores to the alternative method of 8bit/4bit quantization (with similar BOPs). Moreover, when applying a compression factor of 25% (50% on low spatial-sized tensors), we achieve superior results to the quantized alternative. This comparison is presented qualitatively in Appendix A. Finally, in the full 25% compression experiment, we see that even when aggressively compressing the low spatial-sized tensors, we achieve better results than using low-precision standard quantization.

## 6 Conclusion

In this work, we presented a new approach for feature map compression, aiming to reduce the memory bandwidth and computational cost of image-to-image CNNs, where typically the resolution is high. Our approach is based on the classical Haar-wavelet image compression, which has been used for years in standard devices and simple hardware, and in real-time. We save the computational cost by applying $1 \times 1$ convolutions on the shrunk wavelet domain. We show that this approach surpasses aggressive quantization using the same bit rate at minimal additional costs of the transform.

## REPRODUCIBILITY STATEMENT

The code for running and reproducing the experiments reported in this paper is attached as supplementary material.

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

## A  BTS COMPRESSION RESULTS

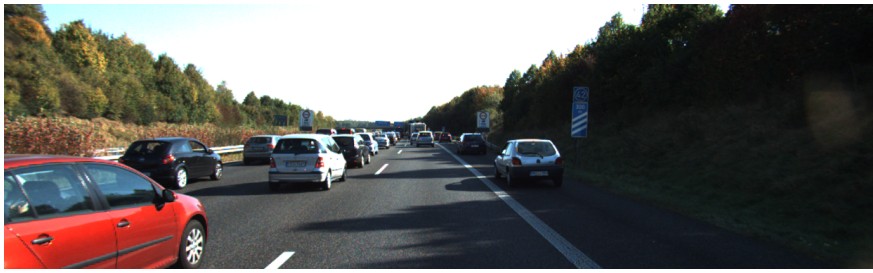

(a) Input Image

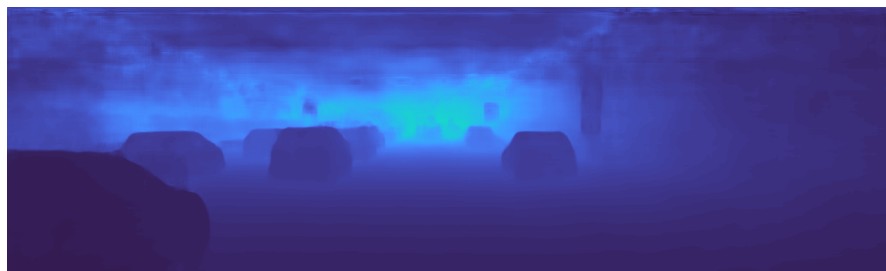

(b) Wavelet Compression 8/8 25% (50% in small tensors)

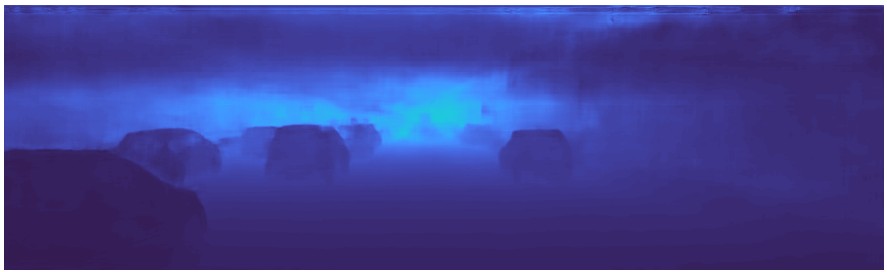

(c) Quantization 8/2 (8/4 in small tensors)

Figure 5: Kitti depth estimation prediction examples. We compare two compressed networks with the same BOP magnitude, the first uses our wavelet compression method and the second uses low-bit quantization.

## B  A NOTE ON $1 \times 1$-CONVOLUTIONS

In the case when a certain CNN use $3 \times 3$ convolution only, one can split it to two convolution, a depthwise-$3 \times 3$ and a $1 \times 1$ (Wang et al., 2017; Vanhoucke, 2014). The depthwise conv involves with $C_{in} \cdot 3 \cdot 3 \cdot N_W \cdot N_H$ MAC operations (assuming no strides), while the $1 \times 1$ conv includes $C_{in} \cdot C_{out} \cdot N_W \cdot N_H$ MAC operations ($C_{out}/9$ times more expensive than depthwise). Meaning, for a large enough $C_{out}$, the $3 \times 3$ convolution has about 8-9 times more MAC operations than the depthwise-$3 \times 3$ convolution and $1 \times 1$ convolution.

Some models, such as the ones referenced in section 4, are defined based on that concept. For example, MobilenetV2 consists of residual blocks that perform $1 \times 1$, depthwise-$3 \times 3$, and an additional $1 \times 1$, and for an image input of size $1024 \times 2048$ (*e.g.* cityscapes), the $1 \times 1$-conv has a MAC count of 18,022M, while the $3 \times 3$-conv has a MAC count of 1,056M (a full breakdown is provided in Table 3).

| Module id | $C_{in}$ | $C_{out}$ | $K$ | Groups | $S$ | Dilation | $H$ | $W$ | MAC |
|---|---|---|---|---|---|---|---|---|---|
| InvRes1 conv1 | 32 | 32 | 3 | 32 | 1 | 1 | 513 | 1025 | 150,552,864 |
| InvRes1 conv2 | 32 | 16 | 1 | 1 | 1 | 1 | 511 | 1023 | 267,649,536 |
| InvRes2 conv1 | 16 | 96 | 1 | 1 | 1 | 1 | 513 | 1025 | 807,667,200 |
| InvRes2 conv2 | 96 | 96 | 3 | 96 | 2 | 1 | 513 | 1025 | 112,914,648 |
| InvRes2 conv3 | 96 | 24 | 1 | 1 | 1 | 1 | 256 | 512 | 301,989,888 |
| InvRes3 conv1 | 24 | 144 | 1 | 1 | 1 | 1 | 258 | 514 | 458,307,072 |
| InvRes3 conv2 | 144 | 144 | 3 | 144 | 1 | 1 | 258 | 514 | 169,869,312 |
| InvRes3 conv3 | 144 | 24 | 1 | 1 | 1 | 1 | 256 | 512 | 452,984,832 |
| InvRes4 conv1 | 24 | 144 | 1 | 1 | 1 | 1 | 258 | 514 | 458,307,072 |
| InvRes4 conv2 | 144 | 144 | 3 | 144 | 2 | 1 | 256 | 514 | 42,467,328 |
| InvRes4 conv3 | 144 | 32 | 1 | 1 | 1 | 1 | 128 | 256 | 150,994,944 |
| InvRes5 conv1 | 32 | 192 | 1 | 1 | 1 | 1 | 258 | 514 | 206,069,760 |
| InvRes5 conv2 | 192 | 192 | 3 | 192 | 1 | 1 | 256 | 514 | 56,623,104 |
| InvRes5 conv3 | 192 | 32 | 1 | 1 | 1 | 1 | 128 | 256 | 201,326,592 |
| InvRes6 conv1 | 32 | 192 | 1 | 1 | 1 | 1 | 258 | 514 | 206,069,760 |
| InvRes6 conv2 | 192 | 192 | 3 | 192 | 1 | 1 | 256 | 514 | 56,623,104 |
| InvRes6 conv3 | 192 | 32 | 1 | 1 | 1 | 1 | 128 | 256 | 201,326,592 |
| InvRes7 conv1 | 32 | 192 | 1 | 1 | 1 | 1 | 258 | 514 | 206,069,760 |
| InvRes7 conv2 | 192 | 192 | 3 | 192 | 2 | 1 | 256 | 514 | 14,155,776 |
| InvRes7 conv3 | 192 | 64 | 1 | 1 | 1 | 1 | 64 | 128 | 100,663,296 |
| InvRes8 conv1 | 64 | 384 | 1 | 1 | 1 | 1 | 66 | 130 | 210,862,080 |
| InvRes8 conv2 | 384 | 384 | 3 | 384 | 1 | 1 | 66 | 130 | 28,311,552 |
| InvRes8 conv3 | 384 | 64 | 1 | 1 | 1 | 1 | 64 | 128 | 201,326,592 |
| InvRes9 conv1 | 64 | 384 | 1 | 1 | 1 | 1 | 66 | 130 | 210,862,080 |
| InvRes9 conv2 | 384 | 384 | 3 | 384 | 1 | 1 | 66 | 130 | 28,311,552 |
| InvRes9 conv3 | 384 | 64 | 1 | 1 | 1 | 1 | 64 | 128 | 201,326,592 |
| InvRes10 conv1 | 64 | 384 | 1 | 1 | 1 | 1 | 66 | 130 | 210,862,080 |
| InvRes10 conv2 | 384 | 384 | 3 | 384 | 1 | 1 | 66 | 130 | 28,311,552 |
| InvRes10 conv3 | 384 | 64 | 1 | 1 | 1 | 1 | 64 | 128 | 201,326,592 |
| InvRes11 conv1 | 64 | 384 | 1 | 1 | 1 | 1 | 66 | 130 | 210,862,080 |
| InvRes11 conv2 | 384 | 384 | 3 | 384 | 1 | 1 | 66 | 130 | 28,311,552 |
| InvRes11 conv3 | 384 | 96 | 1 | 1 | 1 | 1 | 64 | 128 | 301,989,888 |
| InvRes12 conv1 | 96 | 576 | 1 | 1 | 1 | 1 | 66 | 130 | 474,439,680 |
| InvRes12 conv2 | 576 | 576 | 3 | 576 | 1 | 1 | 66 | 130 | 42,467,328 |
| InvRes12 conv3 | 576 | 96 | 1 | 1 | 1 | 1 | 64 | 128 | 452,984,832 |
| InvRes13 conv1 | 96 | 576 | 1 | 1 | 1 | 1 | 66 | 130 | 474,439,680 |
| InvRes13 conv2 | 576 | 576 | 3 | 576 | 1 | 1 | 66 | 130 | 42,467,328 |
| InvRes13 conv3 | 576 | 96 | 1 | 1 | 1 | 1 | 64 | 128 | 452,984,832 |
| InvRes14 conv1 | 96 | 576 | 1 | 1 | 1 | 1 | 66 | 130 | 474,439,680 |
| InvRes14 conv2 | 576 | 576 | 3 | 576 | 1 | 1 | 66 | 130 | 42,467,328 |
| InvRes14 conv3 | 576 | 160 | 1 | 1 | 1 | 1 | 64 | 128 | 754,974,720 |
| InvRes15 conv1 | 160 | 960 | 1 | 1 | 1 | 1 | 66 | 130 | 1,378,713,600 |
| InvRes15 conv2 | 960 | 960 | 3 | 960 | 1 | 2 | 66 | 130 | 70,778,880 |
| InvRes15 conv3 | 960 | 160 | 1 | 1 | 1 | 1 | 64 | 128 | 1,258,291,200 |
| InvRes16 conv1 | 160 | 960 | 1 | 1 | 1 | 1 | 66 | 130 | 1,378,713,600 |
| InvRes16 conv2 | 960 | 960 | 3 | 960 | 1 | 2 | 66 | 130 | 70,778,880 |
| InvRes16 conv3 | 960 | 160 | 1 | 1 | 1 | 1 | 64 | 128 | 1,258,291,200 |
| InvRes17 conv1 | 160 | 960 | 1 | 1 | 1 | 1 | 66 | 130 | 1,378,713,600 |
| InvRes17 conv2 | 960 | 960 | 3 | 960 | 1 | 2 | 66 | 130 | 70,778,880 |
| InvRes17 conv3 | 960 | 320 | 1 | 1 | 1 | 1 | 64 | 128 | 2,516,582,400 |
| Total of $1 \times 1$ | | | | | | | | | 18,022,413,312 |
| Total of $3 \times 3$ | | | | | | | | | 1,056,190,968 |

Table 3: In depth breakdown of MobilenetV2 (as a backbone for deeplabv3+) for a single Cityscapes' image input. $K$ and $S$ refer to the size of the symmetric kernels and strides respectively. The first convolution of the network is omitted, since it is a common practice to avoid quantizing it.

