# OpenReview forum: "Wavelet Feature Maps Compression for Low Bandwidth Convolutional Neural Networks"
_ICLR.cc/2022/Conference — ICLR 2022 Submitted_

### Official Review · Reviewer_pDi1 · 2021-10-20

**Correctness:** 3
**Technical Novelty And Significance:** 1
**Empirical Novelty And Significance:** 2
**Recommendation:** 5
**Confidence:** 4

**Details Of Ethics Concerns:**

I think this paper do not have ethics problems.

**Main Review:**

The strengths of this paper:

1， The authors take the advantage of wavelet to compress the intermediate activation maps

2， The experiments are  sufficient.



The weaknesses of this paper:

1， It seems that only harr wavelet is used. I think Harr Wavelet Feature Maps Compression for Low Bandwidth Convolutional Neural Networks is more appropriate.

2， WCC is designed for 1x1 convolution. However, the kernels of CNNs is usually larger than 1x1.  I think the application spectrum is limited.

3， The method used to compress actitation maps is borrowed from the traditional image compression technique. People is able to take other transform such as Cosine transformation, Fourier transformation to replace the Wavelets transformation. Why authors choose wavelets  here.


**Summary Of The Paper:**

This paper proposes the wavelet compressed convolution for the activation maps of 1x1 convolutions.  The convolution achieves high compression rations and low computational cost at the cost of minimal accuracy loss. The Haar wavelet transform is adopt to conduct hardware friendly  computation.  Extensive experiments proves the performance of this method.

**Summary Of The Review:**

This paper applies the traditional image compression method to compress the activation map and conduct convolution on the transformation domain. This idea is trivial. I do not find the novelty of this paper.

---

> ### Author Response · Authors · 2021-11-19
> **Response to reviewer pDi1**
>
> We thank you for your review and constructive comments. We hope the response below can address your concerns:
>
> 1. Haar in the title: We have no objection to adding Haar to the title if the system allows. It is clear, regardless of the title, that our framework can be applied with other transforms.
>
> 2. WCC for 1X1 convolution: See the general comment on the applicability to $1\times 1$ convs vs. $3\times 3$ convs.
>
> 3. Choice of wavelet: You're absolutely right, and other image compression techniques can be applied to feature map compression. In this work, we specifically chose Haar because it is easy to implement in hardware-efficient code, and has a small linear computational complexity that includes binary operations only, thus allowing us to provide a good computational tradeoff.
>     Applying Fourier transformation is a nice idea, and we believe our framework is generic enough to incorporate new compression methods easily.  FFT will require more high-precision computations than Haar and will have to be applied in patches. The joint shrinkage needs to be proved effective here as well. That is beyond the scope of our paper.
>
> 4. Trivial idea: Many papers present quantization of neural networks for fast inference. The idea of uniform quantization, by itself, is also trivial and old. However, making this work efficiently in a CNN is not so trivial, and many papers had to be written to reach where we are today as a community. Similarly, here we present a better alternative (in our opinion) to uniform quantization that can also be implemented efficiently. We believe that it should render aggressive feature map quantization irrelevant for high-resolution image-to-image tasks. The idea is being seriously considered these days by a company that develops CNN accelerators.
>
> We will be glad to expand if more explanation is needed or more issues arise.

---

### Official Review · Reviewer_C23G · 2021-10-20

**Correctness:** 3
**Technical Novelty And Significance:** 2
**Empirical Novelty And Significance:** 2
**Recommendation:** 5
**Confidence:** 4

**Main Review:**

1. Incomplete experimental results
1-1) Experimental results on more and more various datasets (e.g., COCO), networks (e.g., instance segmentation, super resolution), and configurations (e.g., a variety of precision) should be included.
1-2) Although there are not many low-precision studies targeting segmentation or depth prediction, comparison with SOTA studies must be included. If necessary, it is possible to apply the proposed method to the network for classification and compare this result with various SOTA studies.
1-3) In addition to Haar-Wavelet, there are various wavelet transform techniques such as DWT, so it is necessary to propose a change in performance according to the wavelet technique. Also, it would be good if the performance comparison with a simple down sampling technique is added.

2. Is it possible to extend the proposed method not only to the inference phase but also to the training phase?

3. It is difficult to agree with the argument of this paper because the motivation of this paper is not sufficiently presented. The motivation of this study needs to be clearly presented at the beginning to secure differentiation from many recently published feature-map compression studies.

4. The size of the feature maps and the size of the weight parameters vary depending on whether the layer is located at the front or the deep position, but this paper does not provide a clear consideration for these. Also, weights are often quantized much more aggressively (lower than 8-bits), but the explanation for the relationship between various weight precision and various activation precision is not included.

5. In the process of searching for related research, I found a paper with a similar concept.
"DC-AC: Deep Correlation-based Adaptive Compression of Feature Map Planes in Convolutional Neural Networks," IEEE International Symposium on Circuits and Systems (ISCAS 2021)
This paper also seems to use the concept of image compression for feature-map compression, and therefore, it is necessary to explain the differentiation of the proposed method from this paper.

6. I hope that the description of joint shrinkage can be clearly presented through the figure. In addition, although this paper claims that the goal is to reduce the amount of computation, there is a concern that using shrinkage will eventually lead to an increase in the amount of computation.

7. It seems good to add a summary of the contribution to the last paragraph of the introduction.

8. The composition of Section 4 seems to need some modifications. In other words, since many paragraphs already exist before Section 4-1, it is necessary to present these paragraphs as one new subsection (i.e., Section 4-1) and change the current Section 4-1 to Section 4-2.

9. It is necessary to supplement the explanation of each step in Fig. 2.

10. Why are only 1000 images used as presented in “batch of 1000 images” of Section 5.1? Since there are few cases, it would be good to experiment with all datasets.

11. There are some typos. (e.g., 5page. “wights”  “weights” Table 1. “Wavelet shrikage”  “Wavelet shrinkage”)

**Summary Of The Paper:**

This paper proposed Wavelet Compressed Convolution (WCC), a new approach for activation maps compression for 1x1 convolutions in CNN networks. The proposed method utilizes a hardware-friendly Haar-Wavelet transform, which is widely used in image compression, and performs the convolution on the compressed activation map. This method has high compatibility that can be applied to any 1x1 convolution in the existing network architecture, and shows the outstanding compression efficiency.

**Summary Of The Review:**

The motivation and contribution need to be made clearer. In addition, the experiment part that supports the contribution needs to be thoroughly supplemented. Please address my concerns in "Main Review" through the rebuttal process.

---

> ### Author Response · Authors · 2021-11-19
> **Response to reviewer C23G part 1**
>
> Thank you for your detailed review and constructive comments. We hope the response below can address your concerns:
>
> 1. (1) More experiments: We agree that any paper can be more impressive with additional experiments. While one can argue that we did not present enough variety of networks, we did repeat our experiments with a variety of configurations and achieved consistent results.  (2) Comparisons: Most quantization papers present classification on ImageNet only. There, the images are not very large because the accuracy is not so sensitive to resolution. Furthermore, the most efficient networks perform quite aggressive down-sampling, reaching resolutions of 24X24 or 12X12 in only a few layers. Wavelets do not compress highly downsampled images well - this is a limitation of our proposed work. But in segmentation or depth prediction (or in general, image-to-image tasks), the input image is of higher resolution and is not downsampled that significantly, otherwise important small objects (people, traffic signs) disappear.
> (3) Please see the general comment regarding other types of wavelets.
>
> 2. Training and inference: our method can actually accelerate the training as well since every transform in the forward pass corresponds to an inverse transform in the backward pass and vice-versa. However, since our current implementation is not the fastest we did not mention this. When training on very strong GPUs, sometimes the depthwise convolutions (general ones, not particularly ours) take more time than the $1 \times 1$ convs which involve much more operations but better utilize the GPU power. On low-resource devices (typical for inference), it is the opposite, hence we focus on that case.
>
> 3. We believe that the motivation of this paper is already presented quite well, as suggested by the other reviewers as well. In particular, figure 1 shows the main concept: Haar wavelets are much better at feature compression than aggressive quantization, and we wish to exploit that using the joint shrinkage and the commutativity with $1 \times 1$ convs. We will try to further improve this in the revision.
>
> 4. The size of the feature map indeed varies. Although, when considering large inputs (which is the usual case for image-to-image tasks), the feature map size is sufficiently large for the compression to be effective. For example, that is the reason why in the BTS experiment we used mixed thresholds of 25\% for the larger maps and 50\% for the smaller maps, which are compared to 2 and 4-bit quantization, respectively. These thresholds can be further optimized. Regarding weight quantization, our work is orthogonal to that, which is evident in the segmentation experiment where changing from 8bit to 4bit weights shows a similar degradation in all configurations. This point is indeed important to mention, and we will add a note regarding it in the revised version. That said, this is a complex matter dealt in papers that optimize for dynamic precision (different bits for each layer). This is quite hard to optimize for, and these papers use quite sophisticated methods for the task, e.g., NAS, reinforcement learning. This discussion is beyond the scope of our paper, other than mentioning this in the related work, which will be revised.
>
> 5. Thank you for pointing out this paper; we were not aware of it while working on ours, probably as it was published only recently. The paper exploits the assumption of correlation between channels in the feature maps to compress their residuals for reduced memory footprint. Our work uses the same premise with our joint shrinkage, but the compression is done differently in a way that commutes with $1\times 1$ convolutions. We think that the two methods are complementary to each other and probably can be combined for the low-frequency sub-bands in the wavelet domain that currently require a large portion of the non-zero entries. This can be a fascinating future work. We cited the paper in the revised version.
>
> 6. The joint shrinkage is illustrated using the third step in Fig. 2, for which we explain: "the joint shrinkage into 1D vectors and a single bit-map (or a list of indices),". We revised the text to "the joint shrinkage of the 2D Haar representation into equal-sized 1D vectors and a single bit-map (or a list of indices),". The shrinkage operation, where certain entries are removed, is not dominant in our implementation (time-wise). Also, it is of linear complexity in the number of channels and channel size, while the convolution is of quadratic complexity in the number of channels and linear in the channel size.

---

> > ### Author Response · Authors · 2021-11-19
> > **Response to reviewer C23G part 2**
> >
> > 7.  Summary of contribution: This is the purpose of the last two paragraphs in the introduction. E.g., "We show that the transform...", "We demonstrate the effectiveness...", "We show that using WCC...". We will revise the last paragraph to clearly state "our contribution".
> >
> > 8. Composition of section 4: That is a good idea. We'll modify the composition of section 4 in the revised version.
> >
> > 9. Fig 2: WCC is stated in Eq (7) and surrounding text, and in Fig 2, where further explanations are added to describe the pipeline. We will clarify the caption of Fig 2 refer to its individual parts from the text.
> >
> > 10. 1000 images: This experiment is a brief assessment done over a random batch of images out of Imagenet to test the effects of taking a single layer and compressing it as described. This experiment acts as a view to the inside of the network and can be treated as a proof of concept and preliminary for the other experiments, which present only final results. The experiment can be obtained over the whole ImageNet dataset instead of 1000 random samples (will be updated in the revised version), but we do not expect the result to be different. Regardless, the compression superiority of wavelets is a well-known result. In our everyday lives, images are not compressed using aggressive quantization. They are compressed using transforms.
> >
> > 11. Thank you for spotting the spelling errors; we fixed those errors in the revised version.
> >
> > We will be glad to provide more explanation if needed or more issues arise.

---

### Official Review · Reviewer_QwGk · 2021-10-28

**Correctness:** 3
**Technical Novelty And Significance:** 3
**Empirical Novelty And Significance:** 3
**Recommendation:** 6
**Confidence:** 4

**Main Review:**

## Strengths:
- The paper presents a neat and clever way to apply traditional wavelet-image compression in CNN.
- The qualitative compression assessment appears to be visually convincing and supported by past work on image compression.
- In the method performs decently on a segmentation network on cityscapes as well as a depth estimation network on kitty.
- Supplementary source code: Decompressing the zip file worked. The source code is not documented but readable.
- Working with Haar-wavelets is usually a good start and not unheard of in the machine learning literature:
   - Rinon Gal, Dana Cohen, Amit Bermano, and Daniel Cohen-Or. Swagan: A style-based wavelet-driven
     generative model. arXiv preprint arXiv:2102.06108, 2021.
   - Travis Williams and Robert Li. Wavelet pooling for convolutional neural networks. In International
   Conference on Learning Representations, 2018

   Do so as well for image generation and pooling tasks. Having established the utility of Haar wavelets, more complex wavelets could be explored in future work.


## Open Questions:
### Related work:
- page 3: Williams and Li (2018) write, " For our proposed method, the wavelet basis is the Haar wavelet, mainly for
its even, square subbands. " the pooling filters are, presumably, static in this case?

### Background:
- Filter construction:
It appears the two-dimensional filters in eq. (3) are constructed from 1d-pairs using outer products:
$$ f_{ll} = f_0 \cdot f_0^T , f_{hl} =  f_1 \cdot f_0^T , f_{hl} =  f_0 \cdot f_1^T , f_{hh} =  f_1 \cdot f_1^T ,$$
as described in i. e.
    - [Vyas et al., 2018] Vyas, A., Yu, S., and Paik, J. (2018). Multiscale transforms with application to image processing. Springer.

    The process is currently not obvious. Would authors consider adding a reference or explanation,
    to help readers understand the genesis of the 2d-Haar filters in equation 3?

- Convolution definition:
Why not be more specific just above equation three and speak of a stride two convolution? Doing so would explain why the resolution is
halved for the resulting coefficients $\mathbf{y}_1 , \mathbf{y}_2, \mathbf{y}_3, \mathbf{y}_4 $.

### Wavelet Compressed Convolution
- What are the matrix dimensions in equations six and seven? Are the wavelet and the convolution matrices square or $\mathbb{R}^{n,n}$ in this case?
What about $\mathbf{T}$ and what would the impact of a nonsquare $\mathbf{T}$ on $\mathbf{K}_{1x1}$ be?
Assuming most of the computational work happens there.
Writing this down could help make this section's main points and equation nine easier to understand.

- How are the (tensor ?) dimensions of $\mathbf{x}$ defined? https://www.deeplearningbook.org/contents/notation.html suggests lowercase boldface
should be a vector. A vector makes sense because fast wavelet matrices can be computed by matrix multiplication with an image channel
flattened into a vector. But it seems in equation eight $\sum_j k_{ij} x_j$ sums along a channel dimension?
If that is the case, does $\mathbf{H}\mathbf{x}$ transform the channel dimension in equation seven?

### Computational cost
- How do the input and channel sizes from equations ten, eleven, and twelve relate to the matrices and vectors (?) in equations six to nine?

### Experiments
- For all experiments: of all convolution operations how many are 1x1?
- Segmentation:
    - Does (W / A) mean, weights / activations in Table 1? Perhaps this is something for the caption.
    - The authors may want to consider adding the results of Liu et al. (2021) to table 1.

- Depth estimation:
    - What does BTS stand for, what does it mean?

### Minor remarks:
- page 3: ..., or for utilize ... -> ..., or utilize ...
- page 4: ..., to be trained end to end manner, ... -> .., to be trained in an end to end manner ...
- page 5: wights -> weights
- page 9: Conclusions -> Conclusion

**Summary Of The Paper:**

The paper proposes to use sparsity inducing Haar-Wavelet transforms within 1x1 convolution layers
to reduce memory and compute consumption.
After theoretically motivating their approach, the new method is evaluated.
The evaluation consists of proof of concept as well as two experiments on deep models.
First, a DeeplabV3plus model is compressed and tested on the Cityscapes dataset,
second, a BTS (not explained?) model is compressed and tested on a depth estimation task using
the Kitty dataset. Good performance is observed in both cases.

**Summary Of The Review:**

Overall the paper's main idea is novel and interesting. Many minor issues make the description of the method harder to understand than necessary. I am recommending weak acceptance, for now, assuming that the most important questions will be answered properly during the rebuttal phase. Depending on the answers I am willing to reconsider my recommendation.

---

> ### Author Response · Authors · 2021-11-19
> **Response to reviewer QwGk**
>
> We thank you for your detailed review and constructive comments. We hope the response below can address your concerns:
>
> - First, we will add SWAGAN to the related work on Wavelets in CNNs.
>
> - Regarding the filters: Yes, the filters are static, in both the cited paper and in ours. We do not learn the filters so that we can utilize the fact that their weights are of $\{ 1,-1 \}$, which enables highly efficient hardware implementation (as in binary neural networks - no multiplications are needed). However, experimenting with learned filters, similarly to Wolter et al. (2020), can be an interesting future work.
>
> - Regarding the filter construction: The filters are indeed separable and can also be implemented as the reviewer suggests. We added the citation and mention this in the revision. We use the weights of Eq. 3 to have each level's transform in a single strided conv2d function call in PyTorch. We agree that applying the filters separately results in fewer operations (8n instead of 16n as we present) but at the cost of two PyTorch conv calls, which is effectively slower. Using PyTorch built-in convs, our code is flexible and portable and is actually faster than other packages we've tried. But we agree that a more dedicated implementation using separable filters will be more efficient. We are working on this code following reviewer MK6D comments. Regardless of our implementation, the most important aspect is that the transform can be implemented efficiently on edge devices.
>
> - Regarding the convolution definition: We agree. We updated the paper to include the stride of two in the description.
>
> - Notation and matrix dimension: we follow the notation of the ResNet paper which we found to be a standard notation in the literature. $\mathbf x$ is a hidden layer, or an ``activation'', and is a tensor of size $C_{in}\times N_h \times N_w$ (channels, height, width), or can be viewed as a flattened vector as the reviewer mentions (that is indeed quite common). We refer to the j-th channel as $\mathbf x_j$  which is of size $N_h \times N_w$. $\mathbf T$ and $\mathbf H$ work on each channel separately and in the same way (that is part of the essence here). So, the result of $\mathbf T\mathbf x$ is a tensor where the $j$-th channel is $\mathbf T \mathbf x_j$. The same goes for $\mathbf H$. In terms of dimensions, $\mathbf H \in \mathbb{R}^{N_hN_w\times N_hN_w}$ and $\mathbf T \in \mathbb{R}^{pN_hN_w\times N_hN_w}$ where $p\in [0,1]$ denotes the fraction of elements we keep in the shrinkage.
> We agree that the dimensions can help for some readers, but writing the dimensions of all the actual matrices is not common in deep learning literature and may be cumbersome for others. For example $\mathbf K_{1\times 1}$ is a $1 \times 1$ conv weight, and it operates as the sum in Eq. 8. Its weight dimension is $C_{in}\times C_{out}$. But, the operator (which is a matrix) can be applied on both $\mathbf x$ and $\mathbf T \mathbf H \mathbf x$ (both in Eq. 9) which are of different spatial dimensions. Hence, when looked upon as a matrix, the operator $\mathbf{K}_{1\times 1}$ has different dimensions in each case.
> That said, we will do our best to revise the section so that everything is clear. We think that the reviewer got all of this right in his first read.
>
> - Experiments: Please refer to the general comment regarding the $1 \times 1$ convolutions. We use a MobileNetV2 and ResNext backbones. There, each residual block contains two $1 \times 1$ convolutions and one grouped convolution, which involve with much less computations than the $1 \times 1$. In the decoder, each convolution was split to be separable as well (according to the practice in Chen et. al., 2018), hence, most of the work is done via $1 \times 1$ convolution here as well. A detailed count of all the different convolution layers in MobileNetV2 and their respective operation counts is given in Appendix B in the revised version.
>
> - W/A: yes, this is a common notation and the reviewer is correct. This was further clarified in the revision.
>
> - Liu et. al. in the tables: We made a distinction from Liu et al. since they learn and evaluate the model after resizing images to $256\times 256$, making it meaningless to compare, because they got much lower mIoU than us because of that. That being said, we understand their results can be better compared if added to the table, and we will add them in the revised version.
>
> - BTS is the name of the depth-prediction network we experimented with, Big-to-Small, and we now notice that the name is only mentioned in the authors' code and not in their paper. We clarified this in the revised version.
>
> - Minor remarks: Thank you for spotting the spelling errors; we fixed those in the revised version.
>
> We will be glad to expand if more explanation is needed or more issues arise.

---

> > ### Comment · Reviewer_QwGk · 2021-11-26
> > **Response to authors**
> >
> > Thank you for taking the time to respond to my questions. I saw the methods section now includes additional citations, and the convolution description is more specific.
> >
> > I still feel discussing dimensions in 4 could help improve the link to section 4.1 . The rebuttal tells me that the authors are aware of the dimensions. However, I won't insist since previous work did not always feature such a discussion.
> >
> > I continue to recommend inclusion in the proceedings.

---

### Official Review · Reviewer_mk6D · 2021-11-02

**Correctness:** 2
**Technical Novelty And Significance:** 2
**Empirical Novelty And Significance:** 4
**Recommendation:** 5
**Confidence:** 4

**Main Review:**

Strengths
- The motivation is clear. Feature map compression is heavy in image-to-image modes, and more sensitive than weight quantization in terms of precision (bit). So why not treat feature maps as images and apply image compression techniques there (with wavelet).
- This approach is quite general; it can be applied to any vision models with Conv1x1 layers.
- On semantic segmentation, wavelet compression with shrinkage 25% (effective 2 bit) only only has a small drop in mIOU while direct quantization with 4-bit gives very bad results. From Table 1, wavelet compressed activation leads to a graceful degradation of accuracy while there is a sudden drop of accuracy with direct quantization from 6 bit to 4 bit.

Weakness
- The paper overclaims the effectiveness of the proposed method in the abstract or introduction. It gives the reader the impression that the gain is for general image-to-image tasks. However the improvement is clear for segmentation, but  I don’t see much improvement on depth prediction, even with sophisticated (not-clean) quantization setup. (will be detailed below).  Another misleading message is Figure 1. It basically shows transform coding based image compression is better than quantizing an image in pixel space. However, the MSE for reconstruction of activation maps is not the same thing for the performance of specific tasks. E.g. in Table 1, W8A8 with 0.5 shrinkage (effective W8A4) has similar segmentation mIoU as vanilla W8A6, while in Figure 1,  vanilla A6 achieves similar mIoU to A8 with 0.5 shrinkage (effective A4), instead of A8 with 0.25 shrinkage (effective A2).
- The evaluation of activation compression is through Bits-Operations (BoPs), but no inference time is provided. While BoPs may be one standard metric in model compression literature, it may not directly translate to inference time. E.g. transformers may have lower GMACs than ConvNets, but it does not mean they run faster on GPUs.  The reader would be more convinced if actual runtime on standard hardware (e.g. GPU or NSP) is provided. E.g. the shrinkage operator and zero filling before and after the Conv1x1 may be heterogeneous, and not as efficient as directly applying conv1x1.
- I am not much aligned with the author’s claim on Haar wavelet in image compression (e..g the related sentence in conclusion). About Haar wavelet: 1) it  is not efficient for compression, alternative wavelet with nearly the same complexity achieves significantly better PSNR, 2) it creates very annoying visual artifacts even at moderately low bit rates, similar to blocking, but at many scales, 3) even for very demanding applications, better transforms are used (https://ieeexplore.ieee.org/abstract/document/9190899). I guess the author did not choose a better wavelet as used in JPEG2000, because otherwise the wavelet transforms becomes another 2 layers of separable conv3x3 (or conv5x5). It may go against the original motivation to compress activations after conv1x1.
- The experiment for depth prediction is not clean, with too many models specific quantization setup, e.g. not all conv layers are quantized, it is not clear which layers are actually quantized, and compressed with proposed methods; even when some layers are compressed, certain layers (local planar guidance layers) do not apply wavelet compression; for more aggressive compression, low res feature maps use 50% shrinkage while higher res uses 25%, (25%-50%, 2-4 bit---) etc.
- The result on depth prediction is not good even with those efforts above. From Table 2, the proposed method only achieves on-par accuracy with similar effective bit rate, e.g. vanilla W8A4 is on par with W8A8 plus 50% wavelet shrinkage. It did outperform vanilla A4 on the segmentation task, but again there is a sudden drop from A6 to A4. The last sentence in the abstract is overstatement, because this is the observation only on segmentation tasks, not for general image-to-image tasks.
- One limitation of this paper is that the method only applies to conv 1x1, which is commutative to Haar wavelet transform. The authors claim that conv 1x1 dominates the inference time of major lightweight models, e.g. mobilenet, efficientnet. I would like to see more detailed evidence in the appendix, e.g. some runtime breakdown of conv1x1, conv3x3, etc.  in representative efficient models. Since fully connected layers take lots of compute in big transformer models, I am wondering if it is also suited there?
- Misc
  * It is not clear from the paper how the conv1x1 in the wavelet domain is implemented. Is it  implemented as a fully-connected layer? (in Figure 2) Again what’s the implication for runtime, it would be nice to have a break down of runtime for each stage in the Figure 2, and comparison to the baseline.
  * One thing I am not sure about is whether the baseline quantization method is the sota in model compression literature. I would also suggest the author provide more details on quantization parameters for the experiment in the appendix.
  * I also found the writing in Section 4 is a bit redundant. It would be easier to follow if the author directly provided the equations followed by an explanation.


**Summary Of The Paper:**

The paper proposes to use the Haar wavelet to compress the input  feature maps (activations) of Conv1x1 layers in image-to-image tasks instead of directly quantizing them. The main observation is that wavelet transform is applied channel-wise (separable) and shrinkage of wavelet coefficients is spatial-wise, so they are commutative with Conv1x1. Thus the Conv1x1 layer can be applied after wavelet transform and shrinkage instead of on the original (larger) feature map, leading to savings in the num of bit-operations. The paper claims that the proposed scheme achieves a compression rate comparable to 2-bit on image-to-image translation tasks with only a minor drop in accuracy from 8-bit quantization. However, from the reported results, it is only the case for semantic segmentation, but there is not an obvious gain for depth prediction.

**Summary Of The Review:**

The paper is well motivated and the technical part is straightforward to follow. But the actual gain with this wavelet-based compression of feature maps is not convincing, mainly the added complexity is not reflected in the Bits-Operation metric, the results on experiments are not consistently performing well.

---

> ### Author Response · Authors · 2021-11-19
> **Response to reviewer mk6D**
>
> We thank you for your detailed review and constructive comments. We hope the response below can address your concerns:
>
> - Regarding the over-claimed effectiveness of our method: we do not claim that the reconstruction MSE of the activation maps is exactly correlated to the final evaluation scores of a benchmark task. The MSE is a reasonable measure to guide choices of compression algorithms---e.g., see Banner et. al., NeurIPS 2019, which uses the MSE to determine quantization parameters. The figure demonstrates that the compression of activation maps (not natural images) using the joint Haar compression is much more effective than aggressive quantization. As the reviewer notes, when comparing experiments with similar BOPs our method outperforms the quantization-only setup.
>
> - Regarding the depth prediction results: We agree that the gain in the chosen depth estimation framework is not as prominent as the one for semantic segmentation. The Big-to-Small (BTS) network, from our experience, is much more sensitive to layer modifications than the DeepLabV3 framework, or other simpler frameworks. For example, even applying vanilla quantization was a difficult task. Still, we chose to include these results to demonstrate the usefulness of our method for very low precision (e.g., 2bits). We show that for the same bit-rate setup, our method outperforms the quantization-only setup.
>
> - We agree that actual run-times would be more convincing, but as you mentioned, BOPs are a standard metric for compression. In order to achieve an actual run-time improvement, a low-level implementation of our method is needed, which we are working on right now. We hope it will be ready before the rebuttal period ends, and in that event, we will update you on the results. Regardless of our implementation, it is known that the Haar transform can be very efficient in dedicated hardware, and our aim is to show the potential of this approach in accuracy. This same argument goes for all the quantized operations, which are currently only simulated (in other papers as well).
>
> - Regarding the choice of Haar wavelet: We did not claim the Haar-wavelet is the most efficient qualitatively (in PSNR or other metrics). Instead, we claim it is efficient computationally while being sufficient qualitatively. We will do our best to avoid this confusion in the revised version. See more details in our general comment.
>
> - Regarding the unclean setup: Unfortunately, as explained above, BTS proved somewhat unstable for layer modifications, therefore the non-uniform configurations. Despite those limitations, the benefit from using our method is apparent. In order to present more precise and more significant results, we are currently working on experimenting with another well-known depth-estimation framework. Regarding the current experiments on BTS, we will add more details on the configurations and reasoning behind it in the supplementary material of the revised version.
>
> - Regarding the BTS depth results (second point): Indeed, W8A4 is on par with W8A8 plus 50\%, possibly because some of the error originates from several $3\times 3$ layers that were quantized using 4 bits with no wavelet. However, in lower bit-rates, we do observe a significant advantage, which is the setup for which our method is most effective. Following this research, we are certain that 8bit+0.25\% compression is clearly better than 2-bit quantization, as with natural images.
>
> - Regarding $1\times 1$ convolutions: see the general comment. This paper focuses on CNNs; we agree that a variation of our work for transformers would make a worthy and exciting future work.
>
> - Regarding the implementation: the $1 \times 1$ convolution is implemented using a standard conv1d operation in PyTorch, operating on the compressed channels instead of the channels themselves. The Haar transform is implemented using three rounds of strided depthwise convolutions (for the three levels) with the weights in Eq. (3). This outperformed several packages that we tried. However, indeed this Haar implementation can be further improved for GPU efficiency using a dedicated CUDA implementation. We are working on it, but this way the code will be less portable and flexible. But, the main point is whether the Haar transform can be implemented efficiently on dedicated low-resource hardware, and the answer in our opinion is yes.
>
> - Regarding the quantization being SOTA or not: The training framework of the APoT paper, to the best of our knowledge, is among the best methods for uniform quantization aware training. The work of LSQ presents a similar scheme with somewhat better results on ImageNet, but we could not replicate these results and did not find an online code that does. Regardless, since we use quantization as well, any improvement in this aspect will also improve our method (say, use 6 bits + compression instead of 8).
>
> - In the revised version, we will reconstruct section 4 to include the equations earlier.

---

> > ### Author Response · Authors · 2021-11-28
> > **New results for depth estimation**
> >
> > Dear reviewer mk6D,
> >
> > As we mentioned in the previous reply, we have been working on adopting our method on another depth estimation network, due to the instability in BTS when applying any compression technique.
> > Our chosen model is Monodepth2:
> > Godard C, Mac Aodha O, Firman M, Brostow GJ. Digging into self-supervised monocular depth estimation. In ICCV 2019.
> > We again focus on monocular depth estimation using the KITTI dataset. First, we extended the code-base to support different backbones (such as MobileNetV2) and adapted the depth decoder to use depthwise-separable convolutions, for an extra compression factor with a minimal loss of accuracy (indeed, this did not harm the accuracy).
> > The train/validation split is the default selected by Monodepth2 (based on Zhou et al. ), and we evaluate on the ground truths provided by the KITTI depth benchmark.
> > The compression setup for Monodepth2, compared to BTS, is straightforward. Every layer, except for the first and final three convolutions (which together estimate the depth), is compressed. The 1x1 convolution layers comprise 85\% of the total convolution MACs in the network.
> >
> > | Precision (W / A) | Wavelet shrink | BOPs (B) | AbsRel $\\;$|
> > | Full precision*$\quad\\,$ | None $\quad$$\quad$$\quad$| 3611.7 $\quad$| 0.093 $\quad$|
> > | Full precision$\quad$$\\;\\;$ | None $\quad$$\quad$$\quad\\,$| 1163.6 $\quad$| 0.093 $\quad$|
> > | 8bit / 8bit$\quad$$\quad$$\quad$ | None $\quad$$\quad$$\quad$| 133.6 $\quad$$\\;\\,$| 0.093 $\quad$|
> > | 8bit / 4bit$\quad$$\quad$$\quad$ | None $\quad$$\quad$$\quad$| 99.26 $\quad$$\\;\\,$| 0.097 $\quad$|
> > | 8bit / 8bit$\quad$$\quad$$\quad$ | 50\% $\quad$$\quad$$\quad$$\\;$ | 103.9 $\quad$$\\;\\,$| 0.098 $\quad\\,$|
> > | 8bit / 8bit$\quad$$\quad$$\quad$ | 25\% $\quad$$\quad$$\quad$$\\;$ | 88.5 $\quad$$\quad$| 0.112 $\quad$|
> > | 8bit / 2bit$\quad$$\quad$$\quad$ | None $\quad$$\quad$$\quad$| 82.1 $\quad$$\quad$| 0.268 $\quad$|
> > | 8bit / 8bit$\quad$$\quad$$\quad$ | 12.5\% $\quad$$\quad$$\\;\\;\\;$| 80.8 $\quad$$\quad$| 0.131 $\quad$|
> >
> > We use MobileNetV2 as encoder. The asterisk * denotes the original decoder without applying separable convolutions, while all the rest of the results use the decoder with separable convolutions, which is the first step for compression. Here we see the same trend as in our other results: with our WCC method, we can compress the networks to very low bit-rates while having a quite minimal degradation in the accuracy. In particular, using 25\% and 12.5\% compression, we obtain better accuracy than 2 bits.
> >
> > Best regards,
> > The authors.

---

> > > ### Comment · Reviewer_mk6D · 2021-11-30
> > > **thanks for the rebuttal effort**
> > >
> > > Thank you for the effort to add new results on monodepth2. From the results the setup of W8A4 is still on-par with WCC W8A8 + 50% shrinkage. The general trend for vanilla quantization is that the accuracy degrades drastically below A4 (also the case for segmentation), but with WCC shrinkage, the accuracy drops more gracefully. It seems to me the case for vanilla quantization to A2 is a too weak baseline (basically do not work). Also I am still not convinced about the actual benefit of going to 2 bit without considering the actual added complexity (runtime). I still think the writing in abstract/introduction and Fig 1 are misleading. The discussion about Haar wavelet depends on the implementation and trade-off, it is not easy to reach a conclusion here.
> > > I would like to raise the score to 5 to appreciate the rebuttal effort, but I do not think it is ready for acceptance at this time.

---

> > > > ### Author Response · Authors · 2021-11-30
> > > > **Final response**
> > > >
> > > > Thank you for updating your score and the overall thoughtful discussion. We agree that vanilla A2 quantization is too weak for image-to-image problems. Other papers have shown degradation in accuracy also for classification with A2, so this is not a surprise. We note, however, that currently all the discussion in the literature regarding low-bit quantization is not backed up by actual timings---it is evaluated using measures like bit operations (BOPs). Only 8-bit quantization is really supported at this point, as far as we know. Gaining actual speedups from low-bit quantization requires dedicated hardware, and so does the WCC kernel in our case. We again say that if one wishes to work with less than 4 bits - our approach is more promising than quantization, provided that the hardware performance agrees with the BOPs measure. This assumption is true for both methods considered.

---

### Author Response · Authors · 2021-11-19
**General comments to all reviewers and AC**

We want to thank all the reviewers for their constructive reviews. We will address a few general key points in this comment and answer more specific ones in a personal response.

Some reviewers argued that it might be better to choose different wavelets that are superior to Haar-wavelet. That is true, and, as Reviewer QwGk mentioned, it can be explored in future work. We chose the Haar-wavelet due to its simplicity and computational efficiency, as discussed in Section 1. Other wavelets will require computation using a higher number of bits (most likely 16 or 32) and are not as local as the Haar transform, hence much less hardware friendly. Moreover, one can easily modify our framework to support the use of other wavelets, while considering the additional cost.

We will better explain this point in the revised version. In particular, we will strengthen the point at the bottom of page 1, and in the beginning of the paragraph "Convolution in the wavelet domain" in page 5.

We would also like to address the concerns regarding $1 \times 1$ convolutions. The explanation is two-fold. First, although we did not mention this or implement this, a $3 \times 3$ convolution is usually implemented as a matrix-matrix multiplication using different shifts of the image. Hence the Haar transform can be adapted to transform the shifts as well with a specialized implementation. However, in our opinion, it is not even necessary in the context of network efficiency. In the case when a certain CNN uses $3\times 3$ convolution only, one can factorize it to a set of two convolutions (named separable convolutions), i.e., a depthwise-$3\times 3$ and the other $1\times 1$. The depthwise conv involves with $C_{in}  \cdot 3 \cdot 3 \cdot N_W \cdot N_H$ MAC operations (assuming no strides), while the $1\times 1$ conv includes $C_{in} \cdot C_{out} \cdot N_W \cdot N_H$ MAC operations ($C_{out}/9$ times more expensive than depthwise). Meaning, for a large enough $C_{out}$, the $3 \times 3$ convolution has about 8-9 times more MAC operations than the depthwise-$3 \times 3$ convolution and $1\times 1$ convolution.

Additional information can be found at:
- Min Wang, Baoyuan Liu, and Hassan Foroosh. Factorized convolutional neural networks. In ICCV Workshops, 2017.
- Vincent Vanhoucke. Learning visual representations at scale. ICLR invited talk, 2014.
- Howard et al., Mobilenets: Efficient convolutional neural networks for mobile vision applications. arXiv:1704.04861. 2017.
- Chen et. al., Encoder-decoder with atrous separable convolution for semantic image segmentation, ECCV, 2018.

Some models, such as the ones referenced in Section 4, are defined based on that concept. For example, MobilenetV2 consists of residual blocks that perform $1\times 1$, depthwise-$3\times 3$, and an additional $1\times 1$, and for an image input of size $1024\times 2048$ (e.g., cityscapes), the $1\times 1$-conv has a MAC count of 18,022M, while the $3\times 3$-conv has a MAC count of 1,056M.

The bottom line is (and it fully agrees with our experience): full $3\times 3$ convolutions are redundant, and if one wishes to compress networks, then using separable convolutions is the first step, which should be followed by quantization or other compression methods. We added this explanation to the revised version of the paper together with a breakdown of the operations in each layer of the MobileNetV2 architecture.

We have uploaded a revised version, in which we did our best to address as many of the issues as time permits. Some of the reviewers' comments require a more careful treatment, which we will continue working on past the rebuttal period.

With kind regards,

The authors.

---

### Decision · Program_Chairs · 2022-01-20

**Decision:**

Reject

**Comment:**

The paper receives mixed ratings. All reviewers agree that the paper is well-motivated and the updated draft is clear. However, the experiment results are not fully convincing especially given the added complexity. In addition, it is hard to fully understand where the gain comes from. We hope the reviews can help improve the draft for a strong publication in the future.